# Biologically relevant laminin as chemically defined and fully human platform for human epidermal keratinocyte culture

Monica Suryana Tjin[1], Alvin Wen Choong Chua[2,3], Aida Moreno-Moral [1], Li Yen Chong[1], Po Yin Tang[4], Nathan Peter Harmston[1], Zuhua Cai[1], Enrico Petretto [1], Bien Keem Tan[2,3] & Karl Tryggvason[1,5]

The current expansion of autologous human keratinocytes to resurface severe wound defects still relies on murine feeder layer and calf serum in the cell culture system. Through our characterization efforts of the human skin basement membrane and murine feeder layer 3T3-J2, we identified two biologically relevant recombinant laminins—LN-511 and LN-421- as potential candidates to replace the murine feeder. Herein, we report a completely xeno-free and defined culture system utilizing these laminins which enables robust expansion of adult human skin keratinocytes. We demonstrate that our laminin system is comparable to the 3T3-J2 co-culture system in terms of basal markers' profile, colony-forming efficiency and the ability to form normal stratified epidermal structure in both in vitro and in vivo models. These results show that the proposed system may not only provide safer keratinocyte use in the clinics, but also facilitate the broader use of other cultured human epithelial cells in regenerative medicine.

[1] Program in Cardiovascular and Metabolic Disorders, Duke-NUS Medical School, 8 College Road, Singapore 169857, Singapore. [2] Department of Plastic Reconstructive & Aesthetic Surgery, Singapore General Hospital, 20 College Road, Singapore 169856, Singapore. [3] Skin Bank Unit, Singapore General Hospital, Singapore 169608, Singapore. [4] Department of Anatomical Pathology, Singapore General Hospital, 20 College Road, Singapore 169856, Singapore. [5] Department of Medical Biochemistry and Biophysics, Karolinska Institute, 17771 Stockholm, Sweden. Correspondence and requests for materials should be addressed to A.W.C.C. (email: alvin.chua.w.c@sgh.com.sg) or to K.T. (email: karl.tryggvason@duke-nus.edu.sg)

Recently, De Luca and co-workers[1] demonstrated the possibility to regenerate whole functional epidermis on a 7-year-old child with junctional epidermolysis bullosa, a severe, often lethal, skin blistering disease caused by a mutation in epithelial basement membrane (BM) laminin LN-332 chains[2–5], or in the genes for type XVII collagen, α6, and β4 integrin[6]. The patient's keratinocytes were corrected by gene therapy for a *LAMB3* mutation, and then expanded in vitro on mouse 3T3 fibroblasts, essentially as described by Rheinwald and Green (R&G)[7] to treat large burns[8], and subsequently transplanted to replace the entire skin. Prior to that, Wu et al. showed that engineered epidermal progenitor cells have the potential to function as a gene therapy vehicle to correct diet-induced obesity and diabetes[9]. These two epidermal cell therapy approaches emphasize a huge clinical potential for treatments of lethal or debilitating epidermal diseases. However, the use of the human/mouse xenograft culture system may hinder extensive use of these therapies due to the undefined nature of the system. The 3T3 mouse feeders, bovine serum, and cholera toxin used in the R&G culture system are all classified as ancillary materials (AMs) Risk Tier 4 or high-risk materials according to the U.S. Pharmacopeia USP 29, Section <1043>. From a clinical application standpoint, these AMs when used to produce human cultured skin generate a potential risk of exposing the patient to zoonotic pathogens and immunogenic agents[10]. Furthermore, assessment and removal strategies of these high-risk AMs, in particular for cholera toxin, remain challenging in today's good manufacturing practice systems. Regulatory agencies such as the Food and Drug Administration and the European Medicines Agency currently classify cultured epithelial autografts produced using the R&G method as xenografts and these are approved only for treatment of severe burns (i.e. above 30% total body surface area) or for compassionate use[1,11].

Multiple unsuccessful attempts have been made to develop a human defined cell culture system for expansion of human epidermal keratinocytes (HEKs) in vitro. The attempts include the use of conditioned media from R&G's 3T3-J2 cells[12], human feeder cells, or proteins[13], as well as the use of various protein coatings, such as fibronectin, collagens, and serum-derived vitronectin[14–19]. Although many claims have been made in the literature on having successfully replaced R&G's 3T3 co-culture method, most of the feeder- and xeno-free reported methods still require initial HEKs expansion using a 3T3 feeder layer to obtain the starting HEK progenitor population[15,20,21]. Furthermore, undefined bovine pituitary extract is often used to maintain HEKs in proliferative state and prevent early senescence in feeder-free system[22]. Hence, the development of an effective, reproducible, defined, and fully human system for the culturing of human keratinocytes is still a major unmet clinical need.

In skin, the keratinocytes are positioned on a specialized BM that contains type IV and XVIII collagens, perlecan, and agrin proteoglycans, as well as highly epithelium-specific α3 chain laminin proteins, LN-332, LN-311, and LN-321[23]. In addition to α3 chain laminins, the epithelial BM contains LN-511, a ubiquitous component present in the BMs of almost all tissues, where it forms an adhesion and growth surface for BM-associated cells[23–26]. Pouliot et al.[27] reported that both LN-511 and LN-521 are present in neonatal and adult human skin. They demonstrated that these laminins are potent for α3β1 and α6β4 integrin-mediated adhesion of both neonatal and adult HEKs in vitro. However, it is noteworthy that the LN-511 and LN-521 preparation used in their experiment was derived and purified from a highly crosslinked human placenta extracellular matrix (ECM), containing partially proteolytic protein and also other laminins (e.g. LN-211 and LN-411)[28].

Based on the knowledge about subepithelial BM components, we hypothesized that HEK-associated laminin matrices present in vivo can support the growth of keratinocytes in vitro and replace feeder cells, essentially as we have shown for pluripotent human embryonic stem cells, which can be expanded from single-cell suspension on a pure LN-511 or LN-521 matrix without the use of feeder cells and Rho kinase inhibitors that inhibit apoptosis[29–31].

## Results

**Identification of LN chain expression in human epidermal BM.** We first re-analyzed the ECM components of the human skin BM (laminins in particular) using immunochemistry and RNA-sequencing (RNA-seq) in order to determine which laminin isoform(s) could have potential to support adhesion and propagation of HEKs in vitro. We confirmed that LN-332, LN-511, and LN-521 are the laminins expressed in the sub-epidermal BM (Fig. 1a, b). In addition to these laminins, we also observed high expression level of the laminin β4 chain from RNA-seq data (Fig. 1b). However, currently available laminin β4 chain antibodies did not yield specific staining in the keratinocytes or the epidermal BM (Fig. 1a). Hence, we could not verify its location in human skin. Furthermore, no human recombinant laminin isoforms containing the laminin β4 chain are currently available. In addition to the analysis of laminin expression in human skin, we also examined which laminin chains are mainly expressed in R&G's 3T3-J2 fibroblasts because these cells have been used to this day as feeder layer for HEK culture. We found that the 3T3-J2 fibroblasts mainly express LN-411 and LN-421, with a small proportion of LN-511 and LN-521 (Fig. 1c).

Based on these analyses, we cultured HEKs on LN-511, LN-521, LN-411, LN-421, and LN-332 to compare the adhesion properties of HEKs. The R&G 3T3 co-culture system was used as a positive control, while non-coated plates and LN-111 served as negative controls. All HEK culture on laminins was carried out in an animal-free and defined Keratinocyte Growth Medium-Chemically Defined (KGM-CD) cell culture system without any initial expansion using R&G's 3T3 feeder layer. We observed that LN-511 and LN-521 (but not LN-332) supported freshly isolated adult HEK adhesion and spreading (Fig. 1d). To our surprise, however, LN-411 and LN-421 (the main laminins expressed by murine 3T3-J2 fibroblasts although not present in human skin BM) were also able to support HEK adhesion. We have also tested the laminin coatings against other serum-free defined media such as EpiLife-S7 and CnT-07. However, the use of KGM-CD medium in conjunction with laminins was better than other cell culture media we have tested (unpublished data). There were no significant differences between LN-511 and LN-521, or between LN-411 and LN-421 in supporting HEK adhesion, but we found that LN-511 and LN-421 were able to better sustain HEKs in long-term culture compared to LN-521 and LN-411 (i.e. more terminally differentiated cells were observed in initial culture for LN-521 and LN-411 (Fig. 1d), which resulted in lower population doublings (Supplementary Figure 1). In addition, we also tested possible combinations of laminins (for example: LN-511 + LN-421, LN-511 + LN-332, LN-421 + LN-332, and LN-511 + LN-421 + LN-332), as well as with other BM components such as fibronectin, collagen I, and heparan sulfate proteoglycan. However, there was no additive effect on mixing these components on keratinocyte culture. Therefore, we concluded that LN-511 or LN-421 alone is able to effectively support keratinocyte adhesion and growth.

**In vitro characterization of HEKs grown on LN system.** We compared the growth potential of HEKs cultured using our

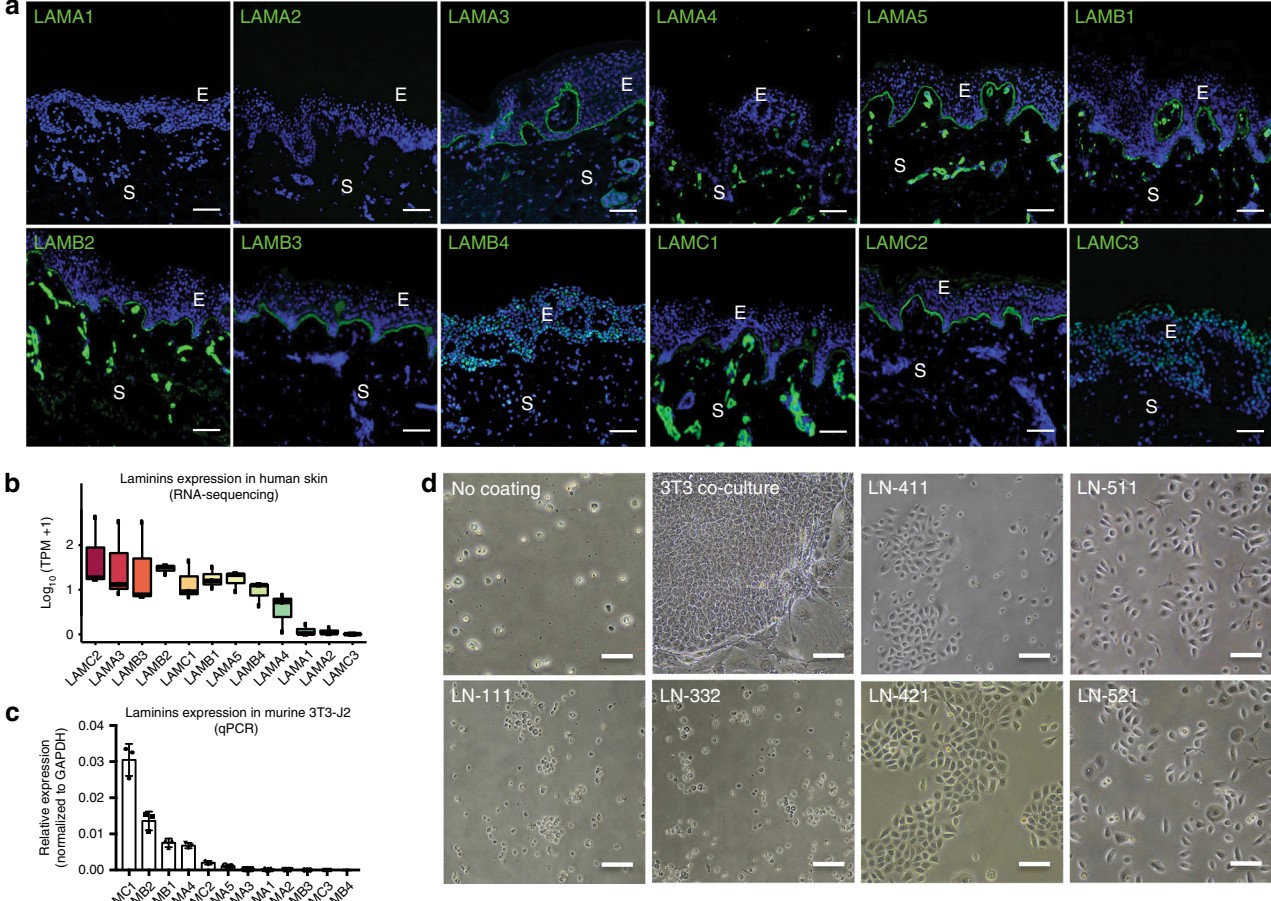

**Fig. 1** Identification of laminin chain expression in human epidermal basement membrane. **a** Immunofluorescent staining of laminin isoforms in skin section. Laminin isoform stainings at human skin basement membrane showed positive for laminin α3, α5, β1, β2, β3, γ1, and γ2 chain isoforms. The laminin α4 chain was not present in the basement membrane but it stained positive on blood vessels. E epidermis, S stroma. Scale bar = 200 μm. **b** Expression of laminin chains in human skin based on RNA-sequencing. **c** mRNA expression level of laminin chains expressed by 3T3-J2 obtained by real-time PCR. Only expression of laminin α4, α5 β1, β2, γ1, and γ2 chains were observed (n = 3, error bar represents means ± s.e.m.) **d** Morphology of human epidermal keratinocytes cultured on various laminin matrices vs. controls (without coating/co-cultured with 3T3-J2 fibroblasts). Cells cultured on plastic only (control), LN-111 (Matrigel laminin), and LN-332 did not survive. Images were taken at day 7 post seeding. Scale bar = 100 μm

laminin system with those generated using 3T3 co-culture system, by maintaining the culture and analyzing their population doublings and colony-forming efficiency. We found that LN-511 and LN-421 were able to support HEK proliferation and colony-forming capabilities comparable to the 3T3 co-culture system (Fig. 2a, b). We also examined the genetic stability of HEKs cultured on LN-511 and LN-421 by karyotyping and found no noticeable translocations or other chromosomal changes (Supplementary Figure 2a).

Subsequently, in order to determine if HEKs cultured using our laminin system were able to retain their basal properties comparable to the 3T3 co-culture control, we performed quantitative PCR (qPCR) on both basal keratinocyte markers *KRT5* and *KRT14*, as well as differentiation markers *KRT1* and *KRT10* over at least 10 passages. In comparison with the co-culture system, keratinocytes on both LN-511 and LN-421 displayed similar trend toward maturation as the passage number increased (Fig. 2c). This demonstrates that our laminin system is robust enough to maintain HEKs in long-term culture, and is comparable to the 3T3 co-culture system.

We next performed fluorescence-activated cell sorting (FACS) analysis to characterize the identity of HEKs cultured on LN-511 or LN-421. We found that at least 90% of the cells that were

freshly cultured on either LN-511 or LN-421 retained their basal markers, suggesting that these cells were keratinocyte progenitors (Fig. 2d, Supplementary Figure 2b). This result was confirmed by immunostaining, showing the expression of basal markers as well as transcription factor p63 (Fig. 2e). *TP63* is known to regulate keratinocyte proliferation and is required for the development and maintenance of keratinocytes in human skin[32].

Additionally, we also carried out organotypic culture to explore if HEKs cultured using our laminin system were able to form a normal stratified epidermal structure in vitro. HEKs cultured on both LN-511 and LN-421 system were able to stratify, forming normal epidermal skin layer with *stratum corneum, stratum granulosum, stratum spongiosum*, and *stratum basale* similar to that of a normal skin and those cultured using 3T3 co-culture system. Comparatively, LN-511 and LN-421 show good maturation of the keratinocytes toward the surface with intact formation of basal layer. We showed the presence of both basal and differentiation marker expression of HEKs cultured in laminin system similar to the 3T3 co-culture system by immunofluorescence staining (Fig. 2f). In addition, we measured the thickness of the stratified keratinocytes in our organotypic culture across all samples and verified that the variations were not significant (Fig. 2g).

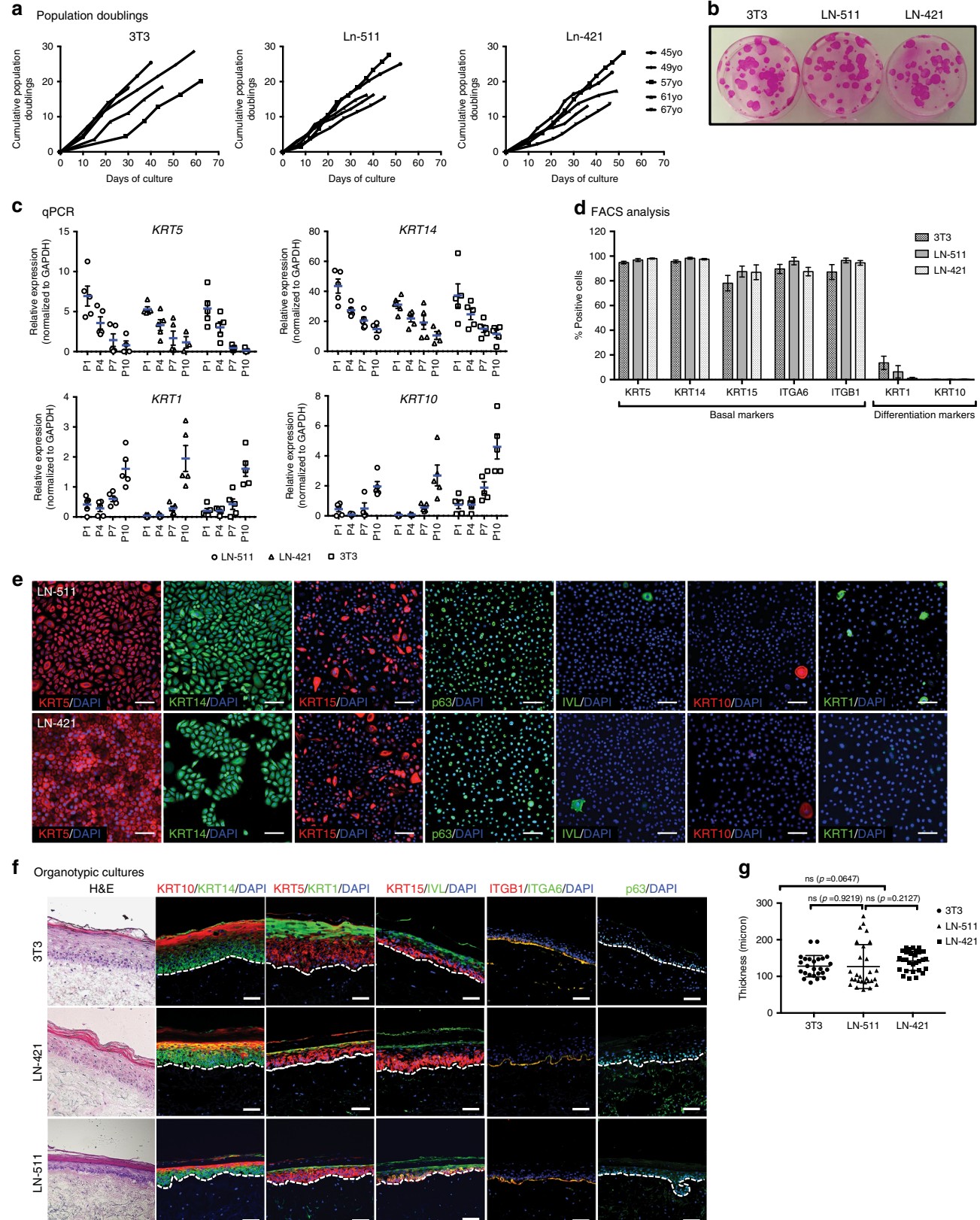

**Transcriptomic profile of HEKs cultured on LN system**. We further profiled and compared the transcriptome of HEKs grown on LN-421, LN-511, and using 3T3 co-culture system by RNA-seq. Basal and differentiation markers showed similar expression levels in the different three culture systems, confirming that the cells grown on the tested platforms were all progenitors (Fig. 3a). Overall, keratinocytes grown on LN-421 and LN-511 showed similar transcriptomic profiles: Spearman's ranked correlation of

**Fig. 2** In vitro characterization of HEKs cultured on laminin system. **a** Growth rate of keratinocytes cultured on LN-511 and LN-421, compared to the conventional R&G's method: co-cultured with 3T3 murine fibroblasts (labeled as 3T3). Population doubling was calculated as PD = 3.32 × log (number of cell harvested/number of cells seeded). **b** Efficiency of colony formation of a representative line of human epidermal keratinocytes from passage 1. **c** Real-time PCR expression profiles of keratinocytes progenitor markers, as well as differentiated cells markers over increasing cell passages. Keratinocytes grown on both LN-511 and LN-421 showed similar trend as the 3T3 co-culture control ($n = 5$, error bar represents means ± s.e.m.). **d** Quantification of progenitor and differentiation markers expression of human epidermal keratinocytes cultured on LN-511, LN-421, and on co-culture with 3T3-J2 feeder cells via FACS analysis ($n = 5$, error bar represents means ± s.e.m.). **e** Immunohistochemical analysis of the expression of keratinocyte markers on LN-511 and LN-421 cultured at passage 1. Scale bar = 100 μm. **f** Epidermal reconstruction on de-epidermalized dermis (DED)/organotypic culture stained with H&E and immunostaining or HEK markers. Scale bar = 100 μm. **g** Thickness measurement of stratified epidermis generated by organotypic culture from laminin (LN-511 and LN-421) vs. 3T3 co-culture system. Dot plot is represented as individual measurement, center line is the means, and whiskers represent s.e.m. ($n = 25$ for 3T3, $n = 30$ for the rest)

0.99, with 60 differentially expressed (DE) genes (false discovery rate, FDR < 0.05), 32 of these genes with a fold change >1.5, and fold changes ranging from 3.4-fold more to 3.1-fold less (Fig. 3b, Supplementary Figure 3a, Supplementary Figure 4, Supplementary Data 1). However, we found large differences when comparing the transcriptome of the keratinocytes grown on the two laminins with the transcriptomic profile of cells grown using the 3T3 co-culture system: Spearman's ranked correlation of 0.94 and 0.95, with 7694 and 7587 DE genes with FDR < 0.05 between keratinocytes grown on LN-511 and LN-421 when compared with the 3T3 co-culture, respectively. Among these 7694 and 7587 DE genes, 5290 and 4894 had a fold change >1.5 with fold changes ranging from 17.5-fold more to 74-fold less, and from 11-fold more to 82.7-fold less in the comparison of HEKs grown on LN-421 and LN-511 compared against the 3T3 co-culture, respectively (Fig. 3c, Supplementary Figure 3b, Supplementary Data 1). In both keratinocytes grown on LN-421 or LN-511, we observed a significant downregulation (FDR < $10^{-6}$) of genes involved in the "epithelial mesenchymal transition" and several pro-inflammatory pathways (e.g. "interferon alpha response", "interferon gamma response", and "TNFA signaling via NFKB"). Pro-inflammatory pathways are known to be upregulated in human skin keratinocytes when co-cultured with fibroblasts[33]. We also observed downregulation of genes involved in several developmental pathways (e.g. "TGF beta signaling" for LN-421 FDR = $9 \times 10^{-6}$ and LN-511 FDR = $1 \times 10^{-4}$, respectively, and "WNT beta catenin signaling" for LN-421 FDR = $4 \times 10^{-4}$ and LN-511 FDR = $1 \times 10^{-2}$, respectively). These keratinocytes also displayed a strong upregulation (FDR < $10^{-6}$) of "MYC targets" and cell cycle (i.e. "E2F targets" and "GM2 checkpoint") among other cellular processes. *MYC* has been shown to regulate keratinocyte adhesion and proliferation[34,35]. Taken together, our data suggest that the transcriptome of the keratinocytes grown on LN-511 and LN-421 are less inflammatory and display more proliferative features than cells grown using the 3T3 graft co-culture system.

**In vivo functional assay of HEKs grown on LN system.** To further validate our in vitro results, we investigated the functionality of these cells in vivo by using a flap transplantation method[36]. This method is suitable for assessing graft survival and for distinguishing between human epidermis generated by the graft from the epidermis of the recipient animal, as it minimizes graft contraction compared to conventional grafting. The graft application and side view of the graft applied using this technique are displayed in Fig. 4a, b. Fourteen days post transplantation, human epidermis generated by keratinocytes cultured on either LN-511, LN-421, or using 3T3 co-cultures were harvested, sectioned and characterized by both hematoxylin/eosin (H&E) staining and immunostaining.

H&E staining revealed that HEKs cultured either on LN-511 or LN-421 were able to generate a fully stratified epidermal layer

in vivo similar to that on the 3T3 co-culture system (Fig. 4c, d). To demonstrate that the origin of the formed epidermis was of human origin, but not from the host, the section was immunostained with antihuman Ku80 nuclear staining antibody (Fig. 4e). Next, immunofluorescence staining of basal and differentiation markers revealed that keratinocytes cultured on either LN-511 or LN-421 were stratified normally in vivo (Fig. 4f). We also observed continuous laminin γ2 chain expression underneath the basal layer of the generated human epidermis on both LN-511 and LN-421 (Fig. 4f, bottom last panels), suggesting that these transplanted HEKs secreted BM proteins (likely epithelial laminin, LN-332) and formed a functional epidermal layer.

**Discussion**

In this study, we have developed a chemically defined and xeno-free method to culture HEKs without feeder layer. We have demonstrated via both in vitro and in vivo characterizations that our laminin system (LN-511 and LN-421) provide robust culture platform to replace the standard H&G's method without compromising the quality of the cells. Based on characterizations that we have performed in this study, we have showed that there were no observable differences between LN-511 and LN-421 in supporting HEK culture. However, we noted keratinocytes cultured on LN-511 or LN-421 displayed slightly different morphology in vitro. While keratinocytes cultured on LN-511 adhered as individual cells, keratinocytes grown on LN-421 formed colony-like morphology reminiscent of keratinocytes co-cultured using a 3T3 feeder layer (Fig. 1d). We also analyzed the migration pattern of the keratinocytes cultured using these two laminins, and observed that keratinocytes on LN-421 moved in a circular/rotational motion similar to HEKs in 3T3 co-culture system (Supplementary Movie 1 and 3), while LN-511 influenced individual keratinocytes to move independently in a directed motion manner (Supplementary Movie 2). Such a rotational motion has been reported to define the "stemness" of keratinocytes[37], even though our transcriptome data showed both laminin systems are nearly identical (Fig. 3b).

To the best of our knowledge, there are no previous reports regarding the function of LN-421 for culturing HEKs. Even though HEKs cultured on LN-511 and LN-421 were morphologically different, we did not observe any significant differences in their characteristics in vivo or in vitro. Our immunostaining, qPCR, and RNA-seq analyses did not reveal the presence of the laminin α4 chain in the sub-epidermal BM, but was expressed by 3T3 cells (Fig. 1a, c).

It is well appreciated that direct contacts with fibroblasts is important for HEKs and adult epithelial cells, in general, to grow rapidly in culture[20]. It has been demonstrated that the use of conditioned medium could not recapitulate the effect that was given by co-culture system with the fibroblasts[12,20]. We hypothesize that a key factor why such direct contact was needed was

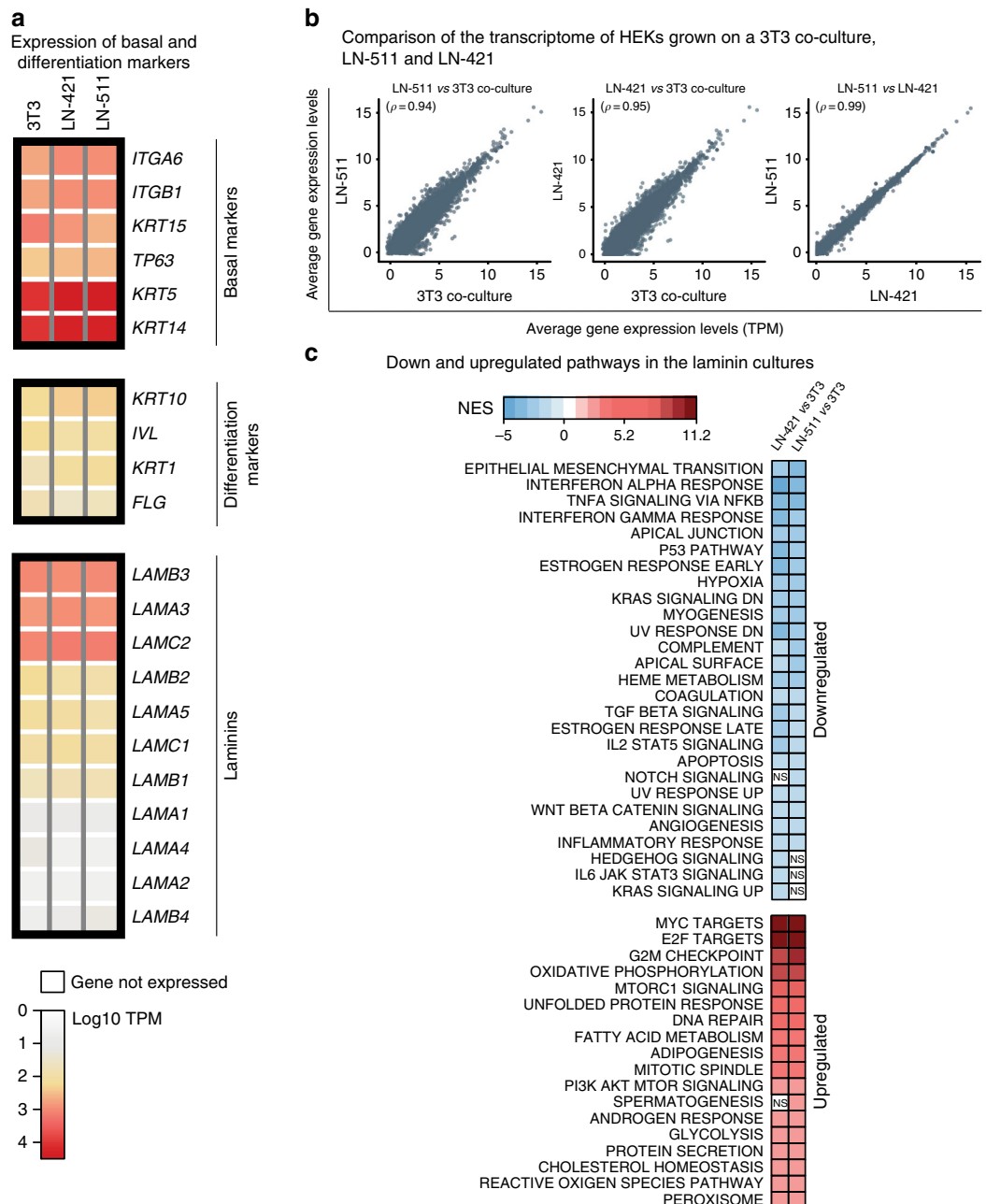

**Fig. 3** Transcriptomic profile of HEKs cultured on laminin system. **a** Heatmap with expression levels of laminin genes and basal and differentiation keratinocyte marker genes (averaged across biological replicates and represented as log10 of transcripts per million, TPM). **b** Correlation between the transcriptome of keratinocytes grown on a 3T3 co-culture, LN-421, and LN-511. Dots represent individual gene expression levels (averaged across biological replicates per culturing method). $\rho$ denotes Spearman's rank correlation coefficient (rho). **c** Functional up- and downregulated processes in keratinocytes growing on LN-421 and on LN-511, when comparing against keratinocytes growing on 3T3-J2 feeder cells. Functional enrichment was computed by Gene Set Enrichment Analysis (GSEA). NES represents the strength of the enrichment and it denotes normalized enrichment score. Nonsignificant processes are indicated with "NS", the rest of processes are all significantly enriched (false discovery rate (FDR) < 0.05). See Methods for more details

the matrix used in the culture coating: the laminin α4 chain expressed by fibroblasts. However, extensive investigations are required in order to validate our speculation, as at this point we are yet to understand the role of this matrix protein on keratinocyte survival and proliferation in culture. No evidence on the role of laminin α4 chain in skin has been shown in the literature, although it was recently reported that the processing of laminin alpha chains (including α4 chain) generates peptides involved in skin wound healing, which possess broad antimicrobial activity

for host defense[38]. Therefore, it will be interesting to further dissect laminin α4 chain function in skin in the future.

Taken together, we have demonstrated the ability to culture functional HEKs in a fully human and chemically defined cell culture system without the need of 3T3 feeder cells using skin-associated laminin matrices. We are convinced that our completely animal-free and chemically defined culture system will not only provide a safer delivery system for patients, but that it also opens up a greater applicability in epithelial stem cell therapy.

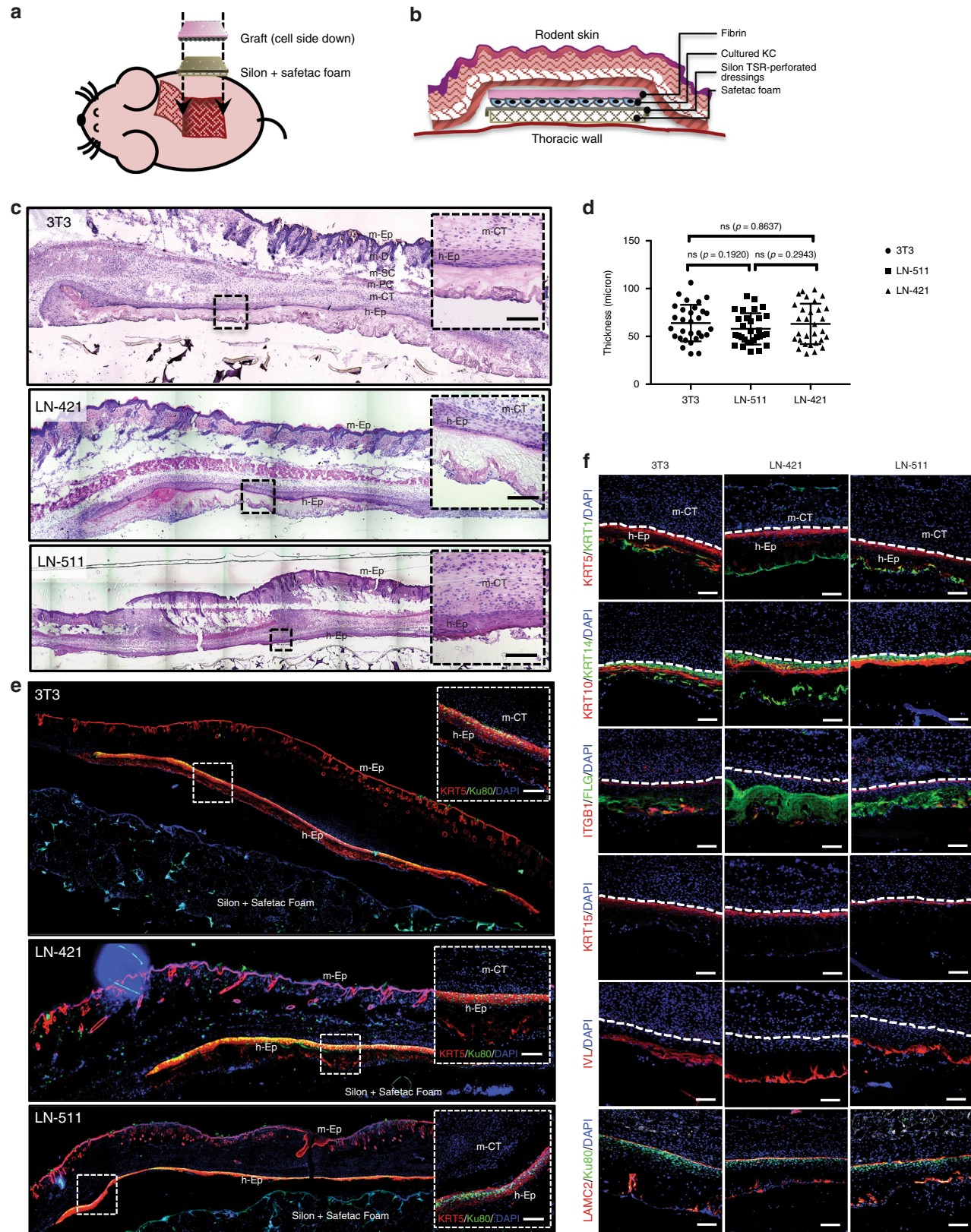

## Methods

**Primary keratinocyte isolation**. HEKs were isolated from surgical waste from plastic surgery operations of healthy subjects, with informed consent from these donors and ethics approval from SingHealth Centralised Institutional Review Board (CIRB#2014/283/D and CIRB#2015/3055). Briefly, a maximum of 4 cm$^2$ tissue was washed in phosphate-buffered saline (PBS) [Lonza] and incubated in 10 mL of 2.5 mg mL$^{-1}$ Dispase II [Roche] in Dulbecco's modified Eagle medium (DMEM) [Gibco] and left overnight at 4 °C. The following day, epidermis was mechanically separated from dermis with fine forceps and incubated in 0.05% trypsin-EDTA solution [Gibco] for 15 min at 37 °C. Upon cellular dissociation,

**Fig. 4** In vivo functional assay of HEKs grown on laminin system. **a** Schematic diagram of the graft site and insertion of silon dressings and graft (with adaptation from Barrandon et al.). **b** Cross section of the completed graft. **c** Human epidermis generated by the flap method, harvested 14 days after grafting consists of the (top to bottom) mouse epidermis (m-Ep), dermis (m-D), subcutaneous tissue (SC), panniculus carnosus (PC), connective tissue (CT), and human epidermis (h-Ep). Frozen sections were stained with hematoxylin and eosin. Scale bar = 100 μm. **d** Thickness measurement of stratified epidermis generated in vivo comparing between laminin culture system (LN-511 and LN-421) vs. 3T3 co-culture system. Dot plot is represented as individual measurement, center line is the means, and whiskers represent s.e.m. (n = 30). **e** Presence of human epidermis is confirmed by human-specific Ku80 staining. Scale bar = 100 μm. **f** Characterization of human epidermal basal and differentiation markers expressed on the developed graft. Note that a continuous laminin γ2 chain-containing layer was observed in between the basal human keratinocytes-mouse connective tissue junction, suggesting that the grafted cells expressed laminin-332—signature expression of functional human keratinocytes. Scale bar = 100 μm

trypsin activity was reduced by diluting the solution with three volumes of fresh DMEM [Gibco]. Keratinocytes were then collected through centrifugation and resuspended in KGM-CD [Lonza]. Donor information is provided as Supplementary Table 1.

**Primary keratinocyte culture.** Six-well tissue culture plates [Corning, Costar] were coated overnight at 4 °C with sterile LN-111, LN-332, LN-411, LN-421, LN-511, or LN-521 [BioLamina AB] at 2.5 μg cm$^{-2}$. Freshly isolated HEKs were seeded on pre-coated plates initially at a density of $9 \times 10^4$ cm$^{-2}$ and cultured in KGM-CD [Lonza] at 37 °C, 10% CO$_2$. For control plates, HEKs were either cultured according to R&G's method[7] or cultured on non-coated plates. Briefly, freshly isolated HEKs ($9 \times 10^4$ cm$^{-2}$) were cultured on a feeder layer of lethally irradiated (60 Gy) 3T3-J2 fibroblasts (gift from late Howard Green's laboratory) in complete FAD medium: DMEM [Gibco] and Ham's F12 [Gibco] media (3:1 ratio) supplemented with 10% fetal bovine serum [Hyclone], 5 μg mL$^{-1}$ insulin [Insulatard®], 0.18 mM adenine [Calbiochem], 0.4 μg mL$^{-1}$ hydrocortisone [Calbiochem], 2 nM triiodothyronine [Sigma], 0.1 nM cholera toxin [Sigma], 10 ng mL$^{-1}$ epidermal growth factor [Upstate], and 100 IU mL$^{-1}$–100 μg mL$^{-1}$ penicillin–streptomycin [Gibco]. Upon confluency, HEK cultures in the serum-free system were trypsinized using TrypLE Select (Gibco Invitrogen) for 8–16 min at 37 °C, while HEK cultures on R&G's control plates were trypsinized with 0.05% trypsin-EDTA [Gibco] for 5 min at 37 °C. Subconfluent primary cultures were serially passaged at $1 \times 10^4$ cells per cm$^2$. The number of cumulative population doublings was calculated using the following formula: PD = (log $N$/$N0$)/log2, where $N$ represents the total number of cells obtained at each passage and $N0$ represents the number of cells plated at the start of the experiment.

**qPCR analysis.** Total RNA from HEK cells at different passages was purified using RNeasy Micro Kit (Qiagen) according to the manufacturer's instructions. The yield was determined by NanoDrop ND-2000 spectrophotometer (NanoDrop Technologies). For quantitative reverse transcription-PCR (RT-PCR) analysis, cDNA was synthesized from 500 ng of total RNA in a 20 μL reaction mixture using TaqMan Reverse Transcription Reagents Kit (Applied BioSystems) according to the manufacturer's instructions. Real-time quantitative RT-PCR was performed with synthesized cDNA in assay mix containing iQ SYBR Green Super mix (BioRad) and primers for genes of interest. GAPDH was used as the normalizing control. Primer sequences are listed in Supplementary Table 2.

**FACS analysis.** Cells were collected at different passages and single-cell suspensions were fixed with Fixation Reagent (Medium A; Life Technologies) for 15 min at room temperature (RT), washed with FACS buffer (0.5% bovine serum albumin and 2 mM EDTA in 1× PBS), blocked with 5% goat serum in FACS buffer, immunostained with primary antibodies in Permeabilization Reagent (Medium B; Life Technologies) for 15 min at RT, detected with secondary antibodies diluted in 1% goat serum in FACS buffer. For fluorophore-conjugated antibodies, fixed cells were incubated with antibodies diluted in Medium B and human FcR blocking reagent (Miltenyi Biotec, 1:50) for 30 min at RT. Stained cells were resuspended in FACS buffer and subjected to FACS analysis (MACSQuant VYB, Miltenyi Biotec). Antibodies and their respective isotype controls used in this study are summarized in Supplementary Table 3. Data were analyzed using MACSQuantify (Miltenyi Biotec) software.

**Colony-forming efficiency assay.** Clonogenic culture was preformed according to Green's method[39]. Briefly, pre-confluent cultures of HEKs were trypsinized from laminin-coated plates or control plates and cell-counted. Two hundred cells, obtained from step-dilution were inoculated into 60 mm tissue culture dishes containing lethally gamma-irradiated 3T3-J2 cells in complete FAD medium. HEKs were cultured for 14 days and fixed in 10% buffered formalin and stained with 1% Rhodamine B (Sigma-Aldrich). In this experiment, HEKs between passages 1 and 3 were used.

**Immunocytochemistry.** Cells, organotypic culture, or animal tissue sections were fixed in 4% paraformaldehyde in 1× PBS for 15 min and permeabilized in PBS/0.1% Triton X-100 solution for 5 min. Samples were washed three times with PBS,

followed by blocking with 5% goat serum for 15 min and incubation with primary antibodies overnight at 4 °C. Following three washes with PBS, the samples were incubated with fluorescence-labeled secondary antibodies for 1 h at RT to visualize the antigens. Following additional three washes with PBS, nuclei of the samples were counterstained and mounted with ProLong™ Gold Antifade Reagent with DAPI (Life Technologies) and visualized under a Leica DMi8 fluorescent microscope.

**Organotypic culture.** Epidermis of glycerol-preserved allogeneic skin (EURO SKIN BANK, EA Beverwijk, Netherland) was removed mechanically after several cycles of snap-freezing and thawing. This de-epidermalized dermis (DED) was then cut into $2 \times 2$ cm squares and the reticular side of the dermis was seeded with $5 \times 10^5$ human dermal fibroblast with the help of a 1 cm diameter stainless steel ring. The next day, each DED was flipped and $2 \times 10^5$ HEK cells that had been grown on either laminin or with R&G systems were seeded separately on individual DED in CFAD medium for 7 days. Subsequently, cultures were lifted to an air–liquid interface for 14 days to stratify. Each sample was then processed for cryosectioning and stained with both H&E and immunostaining. Epidermal thickness measurements was done by taking at least five random locations across each sample and measure the length (height) between *stratum basale* to *stratum granulosum* of the stratified epidermis by using imageJ software. HEKs between passages 1 and 3 were used in organotypic culture experiments.

**Generation of RNA-seq data.** Adult patient-derived keratinocytes were grown separately either in 3T3 co-cultures ($n = 2$), or on LN-421 ($n = 4$) or LN-511 ($n = 3$) coatings. In addition, whole skin was also isolated ($n = 3$). RNA was isolated from either culture plates or whole skin with microRNA purification kit (Norgen Biotek Corporation) according to the manufacturer's guidelines. RNA-seq libraries were prepared using Illumina Tru-Seq Stranded Total RNA with Ribo-Zero Gold kit protocol, according to the manufacturer's instructions (Illumina, San Diego, California, USA). Libraries were validated with an Agilent Bioanalyzer (Agilent Technologies, Palo Alto, CA), diluted, and applied to an Illumina flow cell using the Illumina Cluster Station. Sequencing was performed on Illumina HiSeq2000 sequencer at the Duke-NUS Genome Biology Facility with the paired-end 100 bp read option. The RNA-seq data have been deposited in NCBI's Gene Expression Omnibus[40] (GEO accession number GSE109645).

RNA-seq reads were assessed for quality and aligned to hg38 (Ensembl Gene annotation build 79) using STAR 2.5.2b[41] and quantified using RSEM 1.2.31[42]. We obtained 43 million reads mapped on average in the cultured samples and 113.3 million on average in the samples from whole skin. Gene annotation was retrieved from Ensembl version 79 (hg38) using the R library biomaRt 2.30.0[43]. Ribosomal genes (Ensembl gene biotype "rRNA") and mitochondrial genes were removed (584 genes in total). Small non-coding RNA genes "RN7SL1" and "RN7SL2" were removed as due to their high expression levels, they were outliers in the gene expression distribution. Gene counts were rounded using the R function *round*. A pre-filtering step was added in which we only kept genes with more than one count when summing up across all samples.

**RNA-seq data analysis.** Differential expression analysis was carried out with DESeq2 1.14.1[44], comparing the three culturing methods: 3T3 co-culture, LN-421, and LN-511. DESeq2 was run pairwise using Wald test, collapsing technical replicates, and adjusting for patient effects (i.e. the covariate "Patient_ID" was added in the model). Three pairwise comparisons were carried out, LN-421 samples were compared against 3T3, LN-511 against 3T3, and LN-511 against LN-421. In the DESeq2 results function, the alpha parameter was set to 0.05, the rest of parameters were left as default. Genes were considered significantly DE if Benjamini and Hochberg adjusted p-value < 0.05. The results are visualized in Supplementary Figure 2a and b. The full list of DE genes is presented in Supplementary Data.

Functional enrichment analysis of the differential expression results was performed with Gene Set Enrichment Analysis (GSEA) software 2-2.2.2[45]. All genes included in DESeq2 output were mapped to HGNC symbols and ranked by the corresponding DESeq2 output Wald statistic (i.e. the estimate of the log2 fold change divided by its standard error). GSEA was run assessing overrepresentation of Hallmark gene sets (i.e. coherently expressed gene signatures derived from the

aggregation of groups of annotated gene sets that represent well-defined biological states or processes). Hallmark gene sets were obtained from the Molecular Signatures Database gene sets 5.1. GSEA was run in classic pre-rank mode with 10 000 permutations to assess the FDR. In the GSEA runs, maximum gene set size was set to 5000 and minimum cluster size was set to 10. Gene sets were considered enriched if FDR < 0.05. All results are visualized in Fig. 2d. The rest of the results can be found in Supplementary Data.

To visualize the expression levels of selected genes, transcripts per million (TPM) were pre-filtered by removing lowly expressed genes (i.e. TPMs were summed up across all samples and only genes with TPM higher than one were kept). TPM levels were logged (after adding and offset of 1) and adjusted for patient effects using the function *removeBatchEffect* from the R library limma 3.30.13[46]. Expression levels of laminins, integrins, keratins, basal, and differentiation marker genes were visualized as a heatmap in the three culturing methods (each one averaged across biological replicates, Fig. 2b and Supplementary Figure 3). Expression levels of laminins in the whole skin samples were represented in a boxplot graph (Fig. 1b).

**Karyotyping**. Karyotype analysis of keratinocytes was performed by Cytogenetics lab (Department of Pathology, Singapore General Hospital). Briefly, the cells from LN-511, LN-421, and 3T3 co-culture system plates were treated with colcemid, lysed, and fixed in 3:1 methanol-acetic acid solution. Metaphase spreads were prepared on glass microscope slides, G-banded by brief exposure to trypsin, and stained with Wright solution. A minimum of 20 metaphase spreads were analyzed and documented.

**Preparation of cultured epidermal skin equivalent**. Transparent fibrin mats were prepared in laminar hood using 2 or 5 mL TISSEEL kit (Baxter). Fibrinogen from the kit was diluted two times above the recommended reconstitution using 1.1% sodium chloride solution (NaCl) containing 1 mM of calcium chloride ($CaCl_2$); this solution was subsequently mixed in equal volume with thrombin provided by the kit, diluted to 3 IU mL$^{-1}$ using 1.1% NaCl and 1 mM $CaCl_2$ solution. The above mixed solutions were dispensed uniformly in $10 \times 10$ cm$^2$ dishes, left at room temperature for 10–15 min for complete polymerization, and stored at 4 °C until use. To prevent fibrinolysis during the culture of HEKs on the fibrin mat, aprotinin (Trasylol, Bayer) was added to a final concentration of 150 kIU mL$^{-1}$ in the culture medium at each feeding. For transplantation, fibrin mats were either coated with 2.5 μg cm$^{-2}$ LN-511 or LN-421 and incubated at 4 °C overnight, or seeded with lethally γ-irradiated 3T3-J2 fibroblasts and incubated at 37 °C overnight. HEKs between passages 1 and 3 were then seeded the next day at 10 000 cells per cm$^2$ and grown to confluence. On the day of surgery, all grafts were washed twice with respective serum-free medium (KGM-CD for laminin samples and fresh DMEM for 3T3 co-culture sample).

**Transplantation of human epidermal grafts onto nude mice**. Animal studies were carried out with an approved protocol from SingHealth Institutional Animal Care and Use Committee (IACUC#2016/SHS/1199). Eight- to ten-week-old nude athymic BALB/c nu/nu mice were purchased from Animal Resource Centre (Perth, Western Australia) and used as skin graft recipients. Mice were housed and maintained in SingHealth Experimental Medical Center under specific pathogen-free conditions. All mice were acclimated to their environment for at least 1 week prior the experimental procedure.

On the day of surgery, mice were treated with buprenorphine (1 mg kg$^{-1}$) twice, beginning in the morning (or at least 1 h before the surgery) and another time at the end of the day. In a laminar flow hood, mice were individually anesthetized using 5% isoflurane in a chamber. For maintenance, the mice were subjected to 2% isoflurane via inhalation through a mask for the surgical procedure. Flap procedure was done following the method developed by Barrandon et al.[36]. Briefly, dorsal skin surfaces of the mice were aseptically cleansed twice with alcohol swabs. A rectangular flap of $2 \times 2$ cm$^2$ were incised with scissors and lifted. A sheet of Safetac foam (Mepilex Lite dressing), slightly larger than the flap, was inserted under the animal skin with the sticky side facing down. Subsequently, cultured human epidermal skin equivalent on fibrin (exposed cell surface down) were placed on the Mepilex dressing with two layers of inert Silon-TSR dressing inserted in between to protect the cells. After moisting the graft with a drop of serum-free medium, flap was folded back in place over the graft and incision closed with non-absorbable 6-0 sutures to protect and fix the graft in place.

To harvest the graft, the animal was sacrificed with $CO_2$ inhalation. The graft was then harvested and either fixed in 4% buffered paraformaldehyde and paraffin-embedded, or snap-frozen in liquid nitrogen. Five-micrometer sections were then collected and processed for either H&E staining or immunostaining.

**Statistical analyses of the experimental data**. Data are presented as means ± s.e.m. from three to seven different patient samples. Differences in relative mRNA expression and surface marker protein expression at different time points were assessed with one-way analysis of variance, corrected for multiple comparisons using Tukey's post hoc test. All graphs and statistical analyses were generated by Prism Software 7.0 (GraphPad). Differences were regarded as significant at $p < 0.05$.

## Data availability

The data sets generated during the current study are available from the corresponding authors on reasonable request. The RNA-seq data are accessible from NCBI's Gene Expression Omnibus (GEO accession number GSE109645).

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

## Acknowledgements

We thank Dr. Ma Dongrui for his kind advice and willingness to share his techniques on organotypic culture, and Bryan Edward D. Buenaflor for his help in animal transplantation. This work was supported by NMRC STaR Award grant (NMRC/STaR/0010/2012) to K.T., NMRC grants (NMRC/BNIG/2036/2015) awarded to A.W.C.C., and NMRC grant CBRG15may062 awarded to E.P.

## Author contributions

M.S.T. conducted all in vitro experiments and contributed to the planning, design, analysis of experiments, and writing of the manuscript with the input from all authors. M.S.T. and A.W.C.C. conducted in vivo experiments. A.M.-M. assisted the design of the RNA-sequencing experiments, analyzed the RNA-sequencing data, and wrote the manuscript. L.Y.C. helped with the sample sectioning and stainings. P.Y.T. analyzed sections of organotypic culture and in vivo samples. N.P.H. and E.P. worked with A.M.-M. on the analysis of the RNA-sequencing data. B.K.T. provided all human keratinocytes samples. Z.C. helped M.S.T. in conducting in vitro cell culture. A.W.C.C. conceived the idea, co-lead the study, and contributed to the writing of the manuscript. K.T. supervised the overall project and contributed to the writing of the manuscript.

## Additional information

**Competing interests:** K.T. is a shareholder in BioLamina. The remaining authors declare no competing interests.

