## [Peer Review File · Nature Communications]

Reviewers' comments:

Reviewer #1 (Remarks to the Author):

NCOMMS-18-12856

Biologically Relevant Laminin Matrix as Chemically Defined and Fully Human Platform for Human Epidermal Keratinocyte Culture

This is a very interesting paper. The (xeno-free) culture of human keratinocytes is really a problem.

Successful methods, currently used, still need animal materials such as mouse feeder layer cells, fetal calf serum or bovine pituitary extract. These xenobiotic materials hamper clinical application of these products. Also coating of the culture with basement membrane components has been reported before.

The ultimate goal would be to develop a defined culture method which creates an environment that allows the survival of the keratinocyte stem cell phenotype.

The basement membrane plays an important role in the creation of the keratinocyte stem cell niche and therefore using laminins as a coating is a logic step in the creation of such an environment. However a niche is not created by just one component.

Also from the results can be concluded that indeed the different laminins have different functions. e.g. migration pattern: on LN-511 the cells really migrate and therefore this laminin might be involved in re-epithelialisation, whereas on LN-421 the cells only rotate and seem to proliferate, which would suggest a more 'stemness' function.

Also the reconstructed epidermis in vitro show differences which is not commented on. It would be good to compare it with normal skin. The ultimate aim should be to create an epidermis which is comparable to normal skin epidermis and not to match the 3T3 system.

In order to create the optimal niche for epidermal keratinocytes a combination of different components in my opinion is necessary.

The author studied the expression of different laminins in human skin and showed that different laminins are expressed in the basement membrane. Why did they not study a mixture of these different laminins in their culture system.

I am not convinced that the in vivo model is the best model to study the potential of these cells in regenerating an epidermis. To my opinion an open wound, the future clinical application field, would be better to demonstrate the proof of principle. Even if these wounds contract.

The authors claim that the transcriptome data from the two different laminin coatings are nearly identical; only 60 differentially expressed genes. These 60 differentially expressed genes can really make a big difference in cell behaviour and phenotype. Besides the HEK populations are a mixed population of KC stem cells and transient amplifying cells and possibly more subtypes. Which all have their specific secretome and due to the mixture of cells the different profiles will be faded.

I do not see the added value of supplementary table 3. The fold change is missing. Limiting the table to the most important genes would make it more legible.

In several tables the text is truncated.

The anti-human Ku80 is not mentioned in table S2.

It is not clear which passage cells were used in most experiments.

How many HEK donors were used?

During culture the basal markers decline while differentiation markers increase this means that slowly the stem cells are depleted from the population. This would have tremendous effects on the usefulness in clinic. Can the authors give an estimation on how many doublings are necessary for a 10 X 10 cm wound? What would be the minimal number of cells able to generate an epidermis? What was the passage number of cells used in the organotypic culture?

Reviewer #2 (Remarks to the Author):

The authors attempted to develop xeno-free culture system for human epidermal keratinocytes using laminins, and found that LN-511 and LN-421 enable expansion of adult human skin keratinocytes without mouse 3T3 feeder cells. This study includes some beneficial findings to the researchers in the skin regenerative medicine. Followings are comments from this reviewer.

1) The authors showed LN-511 and LN-421, but not LN-332, LN-521 and LN-411, maintained undifferentiated state of HEKs (Fig 1D). However, the mechanisms were unclear. If the authors evaluate how LN-511 and LN-421 were able to sustain HEKs in long term culture, it would increase the value of this manuscript.

2) In Fig 2A, it has been shown that HEKs cultured on LN-511 or LN-421 had growth potential comparable to HEKs cultured with 3T3 feeder cells, and in Fig. 2F, organotypic culture with HEKs cells, which were grown on LN-511 or LN-421, showed thinner epidermal structure than that with HEK cells cultured with R&G systems. Authors stated "In this study, we presented a chemically defined and xeno-free method to culture HEKs without feeder layer. We have demonstrated via both in vitro and in vivo characterizations that our laminin system (LN-511 and LN-421) provide robust culture platform to replace the standard H&G's method without compromising the quality of the cells." in the first sentence in Discussion. This result raises question about the authors' claim and indicates differences in the differentiation potential in those HEKs.

3) The manuscript lacks donor information of HEKs. Was the study conducted on HEKs from a single donor?

Point-by-Point response to Reviewers' comments on manuscript NCOMMS-18-12856

Reviewer #1 (Remarks to the Author):

1. Biologically Relevant Laminin Matrix as Chemically Defined and Fully Human Platform for Human Epidermal Keratinocyte Culture

Comments: This is a very interesting paper. The culture of human keratinocytes is really a problem. Successful methods, currently used, still need animal materials such as mouse feeder layer cells, fetal calf serum or bovine pituitary extract. These xenobiotic materials hamper clinical application of these products. Also coating of the culture with basement membrane components has been reported before. The ultimate goal would be to develop a defined culture method which creates an environment that allows the survival of the keratinocyte stem cell phenotype. The basement membrane plays an important role in the creation of the keratinocyte stem cell niche and therefore using laminins as a coating is a logic step in the creation of such an environment. However a niche is not created by just one component.

*Response: We thank the reviewer for pointing this out. We agree that the basement membrane niche does not consist of a single component. Other than laminins, subepithelial basement membrane of human skin contains other components. As the reviewer notes, keratinocytes culture on these proteins have been reported previously on serum free medium on other extra-cellular matrix components such as fibronectin and type IV collagen. However, these matrices were tested first after keratinocytes were cultured with 3T3 feeder layer as starting cultures (Woodley et al., 1990). More recently, isolation and culture of keratinocytes populations and their progeny in feeder-free media systems using commercially available serum-free media have been tested with some reported success. **However, none of these systems is knowingly used for autologous keratinocyte culture and grafting in the clinic.** In fact, one of those systems still requires rat-tail type I collagen as coating matrix and all the data obtained were only based on keratinocytes that were passaged twice (up to 3 passages) after primary culture (Lenihan et al., 2014). In contrast, in our experiments, we have managed to culture the human keratinocytes in absolutely xeno-free and serum free media for at least 12 passages. We consider that important and that actually is one key reason for that cells generated with our protocol have been accepted for use in a clinical trial (Singapore General Hospital). .*

We have also tested our keratinocyte cultures on some of these components ourselves (i.e. fibronectin, collagen I, and heparan sulfate proteoglycan) in combination with either LN-511, LN-421, or LN-332. However, the addition of these components did not improve the adhesion and growth of keratinocytes compared to culture on LN-511 or LN-421. A sentence about this has been added to the Results.

References:

Woodley DT, Wynn KC, O'Keefe EJ. Type IV collagen and fibronectin enhance human keratinocyte thymidine incorporation and spreading in the absence of soluble growth factors. J Invest Dermatol. 1990 Jan;94(1):139-43

Lenihan C, Rogers C, Metcalfe AD, Martin YH. The effect of isolation and culture methods on epithelial stem cell populations and their progeny-toward an improved cell expansion protocol for clinical application. Cytotherapy. 2014. Dec;16(12):1750-9

2. Comment: Also from the results can be concluded that indeed the different laminins have different functions. e.g. migration pattern: on LN-511 the cells really migrate and therefore this laminin might be involved in re-epithelialisation, whereas on LN-421 the cells only rotate and seems to proliferate, which for would suggest a more 'stemness' function.

Response: We agree with the comment of the reviewer on the roles of different laminins on keratinocytes. It is a major priority of our current studies on different recombinant human laminins to elucidate the mechanism by which keratinocytes interact with each laminin isoform, but we consider those experiments, which are very extensive and time-consuming ones to be outside the scope of the present article. We hope the reviewer can agree with that and that we do not have to include those experiments in this article.

3. Also the reconstructed epidermis in vitro show differences which is not commented on. It would be good to compare it with normal skin. The ultimate aim should be to create an epidermis which is comparable to normal skin epidermis and not to match the 3T3 system.

Response: We appreciate this suggestion by the reviewer. We have studied this more closely and amended the text of the result section accordingly: "HEKs cultured on both LN-511 and LN-421 system were able to stratify, forming normal epidermal skin layer with stratum corneum, stratum granulosum, stratum spongiosum, and stratum basale similar to that of a normal skin and those cultured using 3T3 co-culture system. Comparatively, LN-511 and LN-421 shows good maturation of the keratinocytes towards the surface with intact formation of basal layer. We showed the presence of both basal and differentiation markers expression of HEKs cultured in laminin system similar to the 3T3 co-culture system by immunofluorescence stainings (Fig. 2f). In addition, we measured the thickness of the stratified keratinocytes in our organotypic culture across all samples and verified that the variations were not significant (Fig. 2g).

Fig. 2g. Thickness measurement of stratified epidermis generated by organotypic culture from laminin (LN-511 and LN-421) vs 3T3 co-culture system. No significant differences were observed across all samples.

4. In order to create the optimal niche for epidermal keratinocytes a combination of different components in my opinion is necessary. The author studied the expression of different laminins in human skin and showed that different laminins are expressed in the basement membrane. Why did they not study a mixture of these different laminins in their culture system.

Response: We have tested various combinations of laminins (i.e. LN-511+LN-332, LN-511+LN-421, LN-421+LN-332, LN-511+LN-111, LN-421+LN-111, LN-332+LN-111, LN-511+LN-421+LN-332). However, we found no additive effect on mixing laminin isoforms such as LN-111 or LN-332 to either LN-511 or LN-421. We

concluded that LN-511 or LN-421 alone can be used to culture keratinocytes effectively. We have amended the result section to reflect these findings.

5. I am not convinced that the in vivo model is the best model to study the potential of these cells in regenerating an epidermis. To my opinion an open wound, the future clinical application field, would be better to demonstrate the proof of principle. Even if these wound contract

Response: We have previously tested our cultured grafts on an open wound in a pre-clinical model (see Fig. below) described by Del Rio and colleagues (Del Rio et al. 2002) with modification. Instead of using the “thick” fibrin matrix described by Del Rio, our group used a much thinner fibrin matrix that is used in our clinic at Singapore General Hospital (Chua et al. 2018) and others (Hirsch et al. 2017) to recapitulate the actual product that will be used in the operating rooms. These data thus can also be used as part of our justification towards clinical trials.

Left: open wound were created on the back of nude mice by removing full thickness of the mouse skin, followed by insertion of human graft (keratinocytes cultured on fibrin matrix, cultured cell side facing up). The graft were protected and sutured together with vaseline gauze and devitalised mouse skin.
Middle: Appearance of the wound 28 days after grafting.
Right: Healed wound by contraction after 48 days.

However with the use of thinner matrices, we found that contraction dominated the healing and hence there was little integration of the cultured human skin onto the mice, even in our control group (n=24, 6 mice per group). This was based on our histological findings as we tried to trace the grafted human cells on the healed wound after animal sacrifice. Additionally, we tried using silicone chambers to prevent contraction, and similar to what was mentioned by Barrandon et al. [Barrandon et al. 1988], we found that the chambers were difficult to use on these delicate mice and worse still, they caused shearing on the grafted cultured sheets. As such, we eventually switched and adopted the “flap grafting” model as described by Barrandon et al. which provided a uniform and reliable platform to investigate the properties of grafted human epidermis.

References:

- Del Rio M, Larcher F, Serrano F, Meana A, Muñoz M, Garcia M, Muñoz E, Martin C, Bernad A, Jorcano JL. A preclinical model for the analysis of genetically modified human skin in vivo. *Hum Gene Ther.* 2002 May 20;13(8):959-68
- Chua AWC, Khoo YC, Truong TTH, Woo E, Tan BK, Chong SJ. From skin allograft coverage to allograft-micrograft sandwich method: A retrospective review of severe burn patients who received conjunctive application of cultured epithelial autografts. *Burns.* 2018 Aug;44(5):1302-1307
- Hirsch T, Rothoefl T, Teig N, Bauer JW, Pellegrini G, De Rosa L, Scaglione D, Reichelt J, Klausegger A, Kneisz D, Romano O, Secone Seconetti A, Contin R, Enzo E, Jurman I, Carulli S, Jacobsen F, Luecke T, Lehnhardt M, Fischer M, Kueckelhaus M, Quaglino D, Morgante M, Biccato S, Bondanza S, De Luca M. Regeneration of the entire human epidermis using transgenic stem cells. *Nature.* 2017 Nov 16;551(7680):327-332
- Barrandon Y, Li V, Green H. New techniques for the grafting of cultured human epidermal cells onto athymic animals. *J Invest Dermatol.* 1988 Oct;91(4):315-8

6. The authors claim that the transcriptome data from the two different laminin

coating are nearly identical; only 60 differentially expressed genes. These 60 differentially expressed genes can really make a big difference in cell behaviour and phenotype. Besides the HEK populations are a mixed population of KC stem cells and transient amplifying cells and possibly more subtypes. Which all have their specific secretome and due to the mixture of cells the different profiles will be faded. Response: We thank the reviewer for raising this point. We purposefully did not add details in the text about the strength of the differences to support this claim. Although there are 60 differentially expressed genes that are significant between HEKs grown on LN-511 and LN-421, the fold changes between these are quite small: only 32 genes have a fold change greater than 1.5, and within this set of 32 genes, the fold changes range from 3.4-fold more to 3.1-fold less (Supplementary Figure 3a). Overall, these differences are minimal for example, in the comparison with of HEKs grown on laminins with HEKs grown on 3T3, we can find 5,290 and 4,894 differentially expressed genes with a fold change greater than 1.5 for HEKs grown on LN-421 and LN-511, respectively. Also in these 5,290 and 4,894 genes, the fold changes range from 17.5-fold more to 74-fold less, and from 11-fold more to 82.7-fold less in the comparison of HEKs grown on LN-421 and LN-511 with 3T3, respectively (Supplementary Figure 3b). We have amended and added a description of some of these details the result section (highlighted by yellow).

7. I do not see the added value of supplementary table 3. The fold change is missing. Limiting the table to the most important genes would make it more legible. Response: We thank the reviewer for this comment. Supplementary Table 3 (Now renamed as Supplementary Table 4) is an excel spreadsheet with the full list of results of the analysis. When preparing the online submission it was converted into a pdf which lead to formatting issues. We have now sorted this out. We would prefer to keep the full list of results as they may be of relevance and interest to the research community and it also makes the study more transparent.

8. In several tables the text is truncated. Response: We have edited the text accordingly.

9. The anti-human Ku80 is not mentioned in table S2. Response: We have added anti-human Ku80 antibody in table S2

9. It is not clear which passage cell were used in most experiments. How many HEK donors were used? Response: New Supplementary Table 1 has been added. We have added information on cell passage used on each experiment accordingly in materials and method section.

Table S1. Skin donor information used in experiments

Donor ID	Age	Gender	Body site
HEK 1	45	F	abdomen
HEK 2	61	M	abdomen
HEK 3	49	F	breast
HEK 4	57	F	breast
HEK 5	67	F	abdomen

10. During culture the basal markers decline while differentiation markers increase this means that slowly the stem cell are depleted from the population. This would have tremendous effects on the usefulness in clinic. Can the authors give an estimation on how many doublings are necessary for a 10 X 10 cm wound? What would be the minimal number of cells able to generate an epidermis? What was the passage number of cells used in the organotypic culture?

Response: We serially passaged our cells (from adult patient) at 1×10^4 cells/cm². In our normal routine with this seeding density, a full 6-well tissue culture plates is confluent within 4-5 days with an average yield of around 4 million cells. For a 10x10 cm area with 1million cells seeding, we estimate the yield to be around 10million cells after 4-5 days culture. For all our organotypic culture, we used keratinocytes between passage 1-3. The information on seeding density is in the Materials and Methods section and we have added information on cell passage used accordingly.

Reviewer #2 (Remarks to the Author):

The authors attempted to develop xeno-free culture system for human epidermal keratinocytes using laminins, and found that LN-511 and LN-421 enable expansion of adult human skin keratinocytes without mouse 3T3 feeder cells. This study includes some beneficial findings to the researchers in the skin regenerative medicine. Followings are comments from this reviewer.

1. The authors showed that LN-511 and LN-421, but not LN-332, LN-521 and LN-411, maintained undifferentiated state of HEKs (Fig 1D). However, the mechanisms were unclear. If the authors evaluate how LN-511 and LN-421 were able to sustain HEKs in long term culture, it would increase the value of this manuscript.

Response: This is an important and fundamental question, which is a major priority of our current studies on how different laminins influence cellular phenotypes and stem cell differentiation. Such work is in progress in our lab separately for different laminins and it is very extensive. To elucidate the mechanism by which keratinocytes interact with each laminin isoform and how different laminin supports HEKs long term culture we will need e.g. integrin-blocking, conditional knockout of laminin alpha4 and alpha5 chain, etc. Given the timeline for this revision, we do not consider it feasible to elucidate the functional role of LN-511 or LN-421 on keratinocytes within the revision of this manuscript. We do think it is important for the keratinocyte and dermatology community to have the present data describing that human keratinocytes can be cultured on specific laminin molecules after 45 years of time when only murine 3T3 cells have worked.

2. In Fig 2A, it has been shown that HEKs cultured on LN-511 or LN-421 had growth potential comparable to HEKs cultured with 3T3 feeder cells, and in Fig. 2F, organotypic culture with HEKs cells, which were grown on LN-511 or LN-421, showed thinner epidermal structure than that with HEK cells cultured with R&G systems. Authors stated "In this study, we presented a chemically defined and xeno-free method to culture HEKs without feeder layer. We have demonstrated via both in vitro and in vivo characterizations that our laminin system (LN-511 and LN-421) provide robust culture platform to replace the standard H&G's method without compromising the quality of the cells." in the first sentence in Discussion. This result raises question about the authors' claim and indicates differences in the differentiation potential in those HEKs.

Response: We acknowledge that there are variations in thickness of the stratified keratinocytes in organotypic culture. We attributed such discrepancy to be possibly

due to the nature of the organotypic experiment *in vitro*, as well as tissue processing. However, we have measured these variations and found that they are not significant (See new Figure 2g)

Fig. 2g. Thickness measurement of stratified epidermis generated by organotypic culture from laminin (LN-511 and LN-421) vs 3T3 co-culture system. No significant differences were observed across all samples.

Additionally, we have done similar measurement on our *in vivo* data as well to support our claim (see new Fig 4d).

Fig. 4d. Thickness measurement of stratified epidermis generated *in vivo* comparing between Laminin culture system (LN-511 and LN-421) vs 3T3 co-culture system. No significant differences were observed across all samples.

3. The manuscript lacks donor information of HEKs. Was the study conducted on HEKs from a single donor?

Response: We have added donor information as Supplementary Table 1.

Table S1. Skin donor information used in experiments

Donor ID	Age	Gender	Body site
HEK 1	45	F	abdomen
HEK 2	61	M	abdomen
HEK 3	49	F	breast
HEK 4	57	F	breast
HEK 5	67	F	abdomen

REVIEWERS' COMMENTS:

Reviewer #1 (Remarks to the Author):

The authors addressed the points of comments satisfactorily

Reviewer #2 (Remarks to the Author):

The authors have now revised the manuscript according to the suggestions and improved their work. I have no further comments on this manuscript.